# Two Novel Mouse Models of Duchenne Muscular Dystrophy with Similar Dmd Exon 51 Frameshift Mutations and Varied Phenotype Severity

**DOI:** 10.3390/ijms26010158

**Published:** 2024-12-27

**Authors:** Iuliia P. Baikova, Leonid A. Ilchuk, Polina D. Safonova, Ekaterina A. Varlamova, Yulia D. Okulova, Marina V. Kubekina, Anna V. Tvorogova, Daria M. Dolmatova, Zanda V. Bakaeva, Evgenia N. Kislukhina, Natalia V. Lizunova, Alexandra V. Bruter, Yulia Yu. Silaeva

**Affiliations:** 1Center for Precision Genome Editing and Genetic Technologies for Biomedicine, Institute of Gene Biology, Russian Academy of Sciences, 119334 Moscow, Russia; baykjulia@gmail.com (I.P.B.); lechuk12@gmail.com (L.A.I.); katerinavarlamova196@gmail.com (E.A.V.); ul.okulova@gmail.com (Y.D.O.); annatvor@mail.ru (A.V.T.); sv.daria.m@gmail.com (D.M.D.); aleabruter@gmail.com (A.V.B.); 2Core Facility Center, Institute of Gene Biology, Russian Academy of Sciences, 119334 Moscow, Russia; pdsafonova@gmail.com; 3Laboratory of Molecular Oncobiology, Institute of Gene Biology, Russian Academy of Sciences, 119334 Moscow, Russia; 4National Medical Research Center of Children’s Health, 119296 Moscow, Russia; zv.bakaeva@gmail.com (Z.V.B.); kislukhina.en@yandex.ru (E.N.K.);; 5Department of Pharmacology, Institute of Pharmacy, I.M. Sechenov First Moscow State Medical University (Sechenov University), 119991 Moscow, Russia

**Keywords:** Duchenne muscular dystrophy, dystrophin, animal model, gene therapy

## Abstract

Duchenne muscular dystrophy (DMD) is a severe X-linked genetic disorder caused by an array of mutations in the dystrophin gene, with the most commonly mutated regions being exons 48–55. One of the several existing approaches to treat DMD is gene therapy, based on alternative splicing and mutant exon skipping. Testing of such therapy requires animal models that carry mutations homologous to those found in human patients. Here, we report the generation of two genetically modified mouse lines, named “insT” and “insG”, with distinct mutations at the same position in exon 51 that lead to a frameshift, presumably causing protein truncation. Hemizygous males of both lines exhibit classical signs of muscular dystrophy in all muscle tissues except for the cardiac tissue. However, pathological changes are more pronounced in one of the lines. Membrane localization of the protein is reduced to the point of absence in one of the lines. Moreover, an increase in full-length isoform mRNA was detected in diaphragms of insG line mice. Although further work is needed to qualify these mutations as sole origins of dissimilarity, both genetically modified mouse lines are suitable models of DMD and can be used to test gene therapy based on alternative splicing.

## 1. Introduction

Duchenne muscular dystrophy (DMD) is a severe X-linked hereditary disorder caused by mutations in various regions of the dystrophin gene (*DMD*) that result in a frameshift and formation of a premature termination codon, leading to either unfunctional truncated protein expression or a complete absence of the full muscular dystrophin isoform (Dp427m) [1]. Absence of dystrophin causes reduced muscle contractility, degeneration, and fibrosis, ultimately leading to the patient’s death due to diaphragm and cardiac muscle dysfunction [2,3,4].

While females bearing a defective *DMD* gene on either of the X chromosomes are usually asymptomatic due to the X inactivation [5], male patients’ life expectancy rarely exceeds 35 years [6].

DMD-causing mutations in the dystrophin gene are located predominantly in either the first few exons or in exons 43–55 [7], with the most commonly mutated region being exons 48–55 [8,9], as found in affected individuals in the Russian population [10].

An idea of developing an effective gene therapy to reduce or relieve the symptoms of DMD has been widely researched [11]. The major challenge is that the coding sequence of the dystrophin gene along with its regulatory elements, being around 14 kb long, is too large to be effectively delivered intracellularly [12]. Thus, a promising approach to DMD gene therapy is to promote alternative splicing of the *DMD* gene aimed at mutant exon skipping and open reading frame recovery, a method being actively studied [13]. Importantly, adequate and representative mouse models carrying common human mutations that replicate patient phenotypes are required to test novel therapies.

Numerous animal models of DMD have been generated over the decades [14], each varying in their ability to replicate symptoms accurately, although rodent models remain the most commonly used ones [13,15,16,17] due to their relatively low research cost, the ease of manipulation and housing, and rapid population production. Moreover, a personalized murine DMD model carrying a previously uncharacterized mutation has been recently reported [18].

One of the most commonly used models is the C57BL/10ScSn mdx mouse line [19]. These animals carry a mutation in exon 23 of the *Dmd* gene that causes complete absence of Dp427 dystrophin [20]. Although exhibiting characteristic histopathological features of DMD, the mdx mouse model is not ideal. These mice have relatively long life expectancy, not much shorter than in wild-type animals, and the disease phenotype is not as severe. The severe dystrophic phenotype occurs after 12–15 months, often accompanied by spontaneous sarcoma [21]. Young mice usually display mild muscle pathology due to highly effective and fast regeneration related to utrophin up-regulation that leads to muscle hypertrophy. There are a few methods to increase the severity of the phenotype and replicate the metabolic impairment seen in patients [22]. Examples of such approaches are change of the genetic background from C57BL/6 to DBA/2 [23,24] or utrophin gene knockout [25,26]. Genetically similar to the C57BL/10ScSn mdx mouse model, the D2.mdx mouse model has a few advantages, such as a low regeneration level and wide fibrotic areas due to latent TGF-β binding protein (LTBP) polymorphism [27]. Unfortunately, the main issue with this model is active osteogenesis that causes calcification development in skeletal muscles and the diaphragm. Despite this, both models are suitable for pharmacological testing.

Therefore, at the moment, the most commonly used models based on C57BL/10ScSn mdx are N-ethyl-N-nitrosourea-induced (ENU) variants—mdx2cv, mdx4cv, and mdx5cv [28]. These mutations in the dystrophin gene lead to splicing disruption and frameshift, making these models valuable for studying dystrophin expression and function. These models are also suitable for gene transfer studies due to the low reversion mutation frequency. In the mdx2cv allele, an alteration in the splice acceptor sequence of intron 42 results in aberrant splicing. The mdx5cv allele has a transversion in exon 10, creating a new splice donor site and a deletion. The mdx4cv allele is characterized by a transition in exon 53, creating a stop codon. These ENU-induced variants show lager variation in fiber size and more severe phenotypes, but their lifespan and common DMD symptoms are similar to those of C57BL/10ScSn mdx [29].

The Dmd mdx–β geo model, which carries the beta-Geo marker downstream of exon 63, and the Mdx52 model with targeted deletion of exon 52 are nowadays usually used in research, especially in studies of non-skeletal forms of dystrophin. They lack all dystrophin isoforms, including Dp260 and Dp140, and demonstrate phenotypes similar to those of mdx models, but also have cardiac and esophagus defects [30,31]. The *Dmd*-null mouse model with deletion of the entire *Dmd* genomic region also has no dystrophin isoforms. *Dmd*-null mice demonstrate severe muscle defects (both hypertrophy and dystrophy), abnormal behavior, male sterility, but, unfortunately, long lifespans and no cardiac dysfunction [15].

Unfortunately, while some of the aforementioned models may not perfectly replicate all aspects of DMD caused by specific patient mutations, but nonetheless are valuable for studying gene therapy approaches, including splicing regulation, it is essential for a model to carry a mutation homologous to mutations found in patients. Here, we report the generation of two mouse lines that carry mutations in exon 51 of the *Dmd* gene—“InsT” and “InsG”; these mutations cause a frameshift and a premature stop codon formation. Although being located at the same spot within the exon, the two mutations cause distinct phenotypes—mice of the InsG line exhibit more severe symptoms, such as a significantly shortened lifespan rarely observed in *Dmd* single mutation models, and more pronounced histopathology. Interestingly, trace amounts of N-terminal dystrophin peptide were found in skeletal muscle and diaphragm histology sections of InsT line animals.

## 2. Results

### 2.1. Two Distinct Modified Lines Were Generated

From a total of 154 zygotes injected with CRISPR-Cas9 mRNA and single guide RNA (sgRNA) mix targeting *Dmd* exon 51 and transplanted, 38 F0 generation pups were born. Editing events were only observed in 10 pups’ samples. Based on the Sanger sequencing data, four mice with apparent frameshift variants (Appendix A) were later bred, of which two produced numerous modified offspring, therefore establishing two mutant lines.

Mutant females were subsequently back-crossed with C57Bl/6 genetic background wild-type mice to eliminate other distant mutations that might have emerged as a result of CRISPR/Cas9 system off-target action up to generation F4, mainly examined in this work.

These two lines are characterized by single nucleotide insertions (Appendix A) at the same position of *Dmd* gene exon 51: NM_007868.6:c.7321_7322insT and NM_007868.6:c.7321_7322insG. Hereafter, these lines and variants will be referred to as simply “insT” and “insG”, respectively.

These point mutations in the coding sequence of the mouse *Dmd* gene, whose full native protein product’s length is 3678 amino acids (AAs), are expected to result in the formation of two truncated variants. The mutations alter the protein starting from the 2441st AA residue. Both variants lead to protein truncation by 1233 AAs, with predicted termination of biosynthesis occurring just three AAs downstream of the mutated site. In the insT variant, the substitution p.S2441F occurs at the site, and in the insG variant—p.S2441C, both followed by a short tripeptide, Cys-Gly-Tyr.

### 2.2. Mutant Lines Display Myodystrophy-Specific Pathology Complex

The main pathology process in dystrophinopathies is gradual necrosis of muscle fibers followed by replacement with connective and adipose tissue, which is clinically manifested by progressive severe muscle weakness with muscle pseudohypertrophy. Pathology includes similar histologic signs: the polygonality of the fiber profiles is lost, the fiber diameters vary, the nuclei begin to migrate to the center, and the invasion of macrophages begins. The presence and severity of signs allows us to indirectly evaluate the course of the disease [32].

Histological analysis of diaphragms of 2-month-old mice of the insG line showed the growth of the perimysium, an increase in the diaphragm thickness (Figure 1B), the presence of extensive foci of replacement with adipose tissue, and significant fibrosis foci (Figure 2C). In the insT line, the thickness of the diaphragm was also increased (Figure 1D), and local foci of fibrosis were presented (Figure 2B), although not as prominently as in the insG line. Both lines display a large number of fibers with centrally localized nuclei (Figure 1C and Figure 2E,F). Histological analysis of the intercostal muscles of the same age males of both lines showed the presence of foci of necrosis at the stage of myophagocytosis, moderate variability in the diameter of muscle fibers, and central nuclei (Figure 1B,C and Figure 2E,F).

The onset of muscle pathology (central nucleation, pseudohypertrophy) in the insT line occurs at no later than 3 weeks of age, whereas in the insG line, the symptoms are already observed at postnatal day 10 (P10) (Appendix A, Table 1). By the end of the first month, the insT mice display an increased centrally nucleated fiber count, presence of atrophic and hypertrophic fibers, and local fibrosis. Then, the pathology reaches a plateau, and myodystrophy ceases to progress. On the other hand, pathology continues to develop in insG mice, ultimately leading to substitution of muscular tissues with connective tissue, and resolves with a lethal outcome at around week 7 — 75% of the population dies off by this age.

Histological analysis of skeletal muscle sections from 2-month-old animals, using gastrocnemius muscles as an example, showed the abnormal variability in the diameter of muscle fibers in insG mice due to a large number of hypertrophic and atrophic muscle fibers and myotubes (Figure 1A). Numerous fibers with central nucleation and fibers with a moderate increase in the number of nuclei were detected in all muscle samples. Replacement of muscle fibers with adipose tissue was not detected. However, multiple foci of necrosis with fragments of immune inflammation and macrophage infiltration were presented (Figure 3E,F). Loss of cross-striations of fibers was detected in the damaged areas and in their close proximity (Figure 3F). The insT line also showed the presence of centrally nucleated fibers and variability in muscle fiber diameter, but no wide areas of necrosis were found. Local fibrosis and isolated deteriorating muscle fibers were detected in some samples (Figure 3D).

One of the DMD characteristic pathologies is muscle fiber diameter variability [33]. A large number of fibers with a diameter below average is indicative of an active regeneration process with new myotube formation (muscle fiber precursors) and/or necrosis and degeneration that are accompanied by muscle fiber atrophy and shrinkage. Deviation of muscle fiber diameters toward a larger than average size suggests the presence of hypertrophic fibers. We used the minimal Feret’s diameter as the key morphometric parameter to measure muscle fiber cross-sectional size [34] along with its coefficient of variation, revealing significant differences between model and wild-type animals. Summarized and statistically analyzed data are presented in Table 2 and Figure 1A,B.

According to our data, the majority of fibers in skeletal and intercostal muscles of wild-type animals have a diameter of 20 to 30 μm (66.3 ± 2.8% and 52.0 ± 0.9% of total fiber number, respectively). In the insT line, fibers 20 to 30 μm wide also represented the majority of fibers (56.2 ± 1.2% and 27.4 ± 1.5%), whereas in insG line animals, most of the muscle fibers were smaller, ranging in diameter from 10 to 20 μm (32.7 ± 1.2% and 39.2 ± 1.6%) (Figure 1A).

No differences between any measurements of quantitative markers were observed between F2, F3, and F4 generations (*p* > 0.5, ANOVA).

The presence of adipose replacement areas in the diaphragms of mutant animals that we observed was one of the signs of severe Duchenne muscular dystrophy, so it was decided to use Regaud’s and Haematoxylin-Basic Fuchsin-picric acid (HBFP) stainings to assess the condition of the cardiac tissue. The presence of black and/or dark brown stained areas following Regaud’s staining and red areas after HBFP would indicate early myocardial damage. However, no signs of pathology such as early myocardial ischemia or necrosis in the structure of the myocardium were observed (Figure 4).

The summarized data on histopathology seen in insT and insG lines are presented in Table 3.

### 2.3. Full Dystrophin Isoform Is Lost in Mutant Animal Muscles

To unravel what determines the phenotypic difference between generated lines, we performed an IHC staining of skeletal muscles and diaphragms of wild-type, insG, and insT lines with fluorescently labeled antibodies against dystrophin C-terminus (Figure 5, top and middle panels). Sections of wild-type skeletal muscles and diaphragm displayed prominent fluorescent signals corresponding to dystrophin protein, while both mutant lines displayed minimal fluorescence at a level and in a pattern comparable to secondary antibody-only control (Appendix A). At the same time, staining with fluorescently tagged antibody that recognizes dystrophin N-terminal motif demonstrated the presence of trace amounts of N-terminal dystrophin peptide in skeletal muscles and diaphragm of insT line animals (Figure 5, middle and bottom panels).

This could be explained by residual anchoring of the truncated muscular form of dystrophin to the cellular membrane by actin bound to other membrane proteins, or independently, with its rod domains [35], and is frequently observed in carboxy terminal truncated mutants [36].

To further investigate the phenomenon, expression analysis was performed.

### 2.4. Dystrophin Expression Is Affected at mRNA Level

DMD protein has numerous isoforms, some of which are regarded as tissue-specific [1]. We assessed the expression levels of dystrophin isoforms, namely Dp427m (main muscular isoform), Dp427c (cortical), and Dp71 (ubiquitous) using real-time PCR. Measurements were performed on total RNA from whole brains, skeletal (gastrocnemius) muscles, and diaphragms. Two males from the insT line, three from the insG, and four sibling wild-type animals were examined.

*Dmd* is localized on the X chromosome, which implies that male nonsense mutant mice would only express truncated forms of Dp427, Dp260, and Dp140 (should there be no nonsense-mediated decay or exon skipping taking place) and full unaffected forms of Dp116 (Schwann cell-specific) and Dp71, since their promoters are genomically located downstream of the mutation site [1].

The results suggest that Dp427c isoform expression is substantially reduced in the brain of mutant lines (Figure 1F) and as expected, is absent in other tissues.

The wild-type Dp427m mRNA level displays high variance, but it seems it is hardly affected in skeletal muscles of mutant lines. However, an increase in its expression was detected in diaphragm samples from the insG line (*p* = 0.04, Kruskal–Wallis).

Dp71 expression was not perceivably affected in any group. We observed slight elevation in its expression, although barely reaching significance, in diaphragms and skeletal muscles of insG line mice as compared to other groups (*p* = 0.0497, *t*-test or *p*~0.1, Kruskal–Wallis), but likely negligible in its biological effects (a 3.5-fold mean elevation). It can also explain the remaining fluorescence in C-terminally stained muscle sections (Figure 5), since Dp71 expression is not suppressed, and this isoform possesses the domains needed to bind to membrane complexes on its own, possibly interfering with independently anchored truncated variants and scarce basal read-through variants of full isoform [37].

### 2.5. Survival Analysis

The lifespan of male mice from both lines was evaluated. We found that despite the mutations in both lines being located in the same region, both resulting in a frameshift, the lifespan of hemizygous males differed drastically: the lifespan of males from the insT line was somewhat shorter than wild-type, whereas the insG population had a significantly shortened lifespan (*p* = 0.0005, Kaplan–Meier estimator) and gets dramatically reduced within a month (Figure 1E).

## 3. Discussion

DMD pathology development is induced by the absence or deficiency of functional full-length dystrophin protein. Dystrophin plays a key role in stabilizing striated muscle fibers during contractions by linking the cytoskeleton to the extracellular matrix via dystrophin-associated protein complex (DAPC). Mechanical stress during the fiber contraction in the absence of sufficient dystrophin protein levels leads to an increase in intracellular Ca^2+^ levels, causing protein degradation, mitochondrial dysfunction, and necrosis [1]. Necrosis regions are then infiltrated by macrophages, accompanied by a local increase in pro-inflammatory cytokines, which in turn leads to the formation of chronic inflammation areas. Regions of chronic inflammation undergo constant processes of muscle fiber degeneration and tissue regeneration by myosatellite cells. At the late stages of pathology development, regeneration slows down or stops completely, leading to fibrosis or adipose replacement [1].

Although disease pathogenesis is well understood, the complexity of the underlying processes makes the histological assessment complicated. Histopathology analysis of muscle tissue includes the assessment of its general structure with H&E staining. Healthy muscle fibers are of similar size and shape and have peripherally located nuclei and thin connective tissue layers in the perimysium and endomysium. Pathology, on the other hand, is characterized by the following features, depending on the severity: presence of necrotic fibers with fragmented sarcoplasm, often surrounded by basophilic stained cells; areas of macrophage infiltration; presence of small round muscle fibers at the early stages of regeneration, fibers of different maturity with centrally located nuclei, mature hypertrophic fibers, endomysium and perimysium connective tissue expansion, fibrosis, and regions of lipomatosis.

According to our data, the histopathology of the two generated mouse lines, insG and insT, recapitulates the phenotype seen in DMD human patients. All samples demonstrated classical myodystrophy signs at the necrosis/regeneration stage, including significantly variable muscle fiber diameter, the presence of macrophage infiltration areas, muscle fibers of various levels of maturity with centrally located nuclei, extensive perimysium and endomysium connective tissue proliferation, presence of fibrosis, and adipose replacement regions. Interestingly, the insG line has a more pronounced phenotype (Table 3), as well as a shorter lifespan. Aside from the direct effects of the mutant isoform developing in liveborn mice, some kind of pre-birth selection resulting in a bias toward a more viable and healthier male insT offspring population might have taken place, although this assumption has not been verified and is subject to further investigation.

DMD is a severe and relatively common disorder with a reported incidence of 1:3500–1:6000 among boys [1]. Extensive research into the disease pathology and molecular mechanisms has led to the generation of several animal models that carry mutations in the dystrophin gene. The ideal disease model should not only exhibit a representative phenotype involving all normally affected tissues, but also carry a mutation homologous to a common mutation found in affected individuals. Generated models, mostly mouse lines, recapitulate disease phenotype, although not fully, limiting their use in pathology research and/or therapy discovery and testing. The most well-known DMD mouse models are listed in Table 4. Well-studied Mdx and *Dmd*-null lines can be used both for research purposes and the development of drugs aimed at muscle tissue support. Unfortunately, both are unsuitable for testing drugs against cardiomyopathy due to its mild levels in these animals. The Mdx52 mouse model [17] carries a mutation homologous to common human mutations found in patients, but it also exhibits calcification in limb muscles and diaphragm not observed in humans. Moreover, this line is characterized by the absence of macrophage infiltration and adipose replacement, seen in affected individuals. The double knockout *mdx-utrn^−/−^* model demonstrates a severe DMD phenotype but has limited applicability in therapy development and testing due to utrophin knockout, which will make it difficult to assess its effectiveness in compensating for the missing dystrophin [38]. The insT and insG lines reported in this work carry mutations homologous to a common human mutation and develop a phenotype similar to that observed in patients. Both mouse lines have characteristic histopathology in skeletal muscles and diaphragm, making them useful in muscle pathology studies. On the other hand, these models do not show any cardiomyopathy signs. Many DMD models that exhibit heart pathology do not manifest it at earlier stages of life (Table 4), or are based on substantial loss of DMD protein and/or deficiency of proteins that can compensate for its loss (e.g., Utrophin). Our models were characterized at the age of 2 to 3 months and only bear a single nucleotide insertion at exon 51, which corresponds to the distal part of the central rod domain. Describing heart pathology of the lines generated in our work at later stages of life could be of high interest, but very few of the insG specimens live to the age of more than 4 months, which makes the insT line more preferable for cardiac pathology investigation.

The insG myodystrophy model generated in this work demonstrates the phenotype seen in mild to severe DMD cases; the mutation is homologous to the most common dystrophin mutation found in the human population and thus can be used for the development and testing of both pharmaceutical agents and gene therapy designed for alternative splicing regulation.

While the mutants were back-crossed with wild-type mice to “tidy up” the genetic background of their mutations, no disease-modifying mutations were directly excluded or identified. Although both models were generated in the same manner, considering the substantially shorter lifespan of the insG model compared to that of the insT model, both lines must be more deeply characterized molecularly to find the fundamental basis for such a severity of phenotype and phenotype distinction. Additionally, functional characterization of the models would be of high importance to declaring them suitable for conventional symptom amelioration therapy research and is an indispensable condition for assessing functional restoration upon therapy. Further research is required to elucidate the molecular mechanisms of pathogenesis in the mouse lines generated in this work, as well as assessing the effect of the introduced mutations on the animals’ behavior and motor functions.

## 4. Materials and Methods

### 4.1. Animals

Mice of the C57BL/6, F1 CBA × C57BL/6, and outbred CD1 lines, used in this study, were bred in the animal facility of the Institute of Gene Biology, Russian Academy of Sciences (IGB RAS). The animals were given ad libitum access to water and feed. The dark/light cycle was 13/11 h, air temperature—23 ± 1 °C, humidity—42 ± 5%.

To generate genetically modified animals, prepubescent F1 CBA × C57BL/6 female mice weighing 12–13 g were used as zygote donors. Males of 6–8 weeks of age from the F1 CBA × C57BL/6 line were used as breeders. Outbred CD1 females, 8–10 weeks old, were used as surrogate mothers for embryo transfer after genome editing system delivery to zygotes.

The two lines of genetically modified mice generated in this study were housed and bred separately in the facility.

All experiments involving animals were carried out in adherence with local regulations and approved by the IGB RAS bioethics committee.

### 4.2. Cas9 mRNA and sgRNA Preparation

To produce Cas9 mRNA, the pET28a/Cas9-Cys (addgene #53261) plasmid was linearized with XhoI enzyme to generate a fragment terminating at the end of Cas9 ORF. The fragment was purified by isopropanol precipitation and used as a matrix for in vitro transcription performed with the HiScribe™ T7 kit (New England Biolabs, Ipswich, MA, USA) that allows for capping and polyadenylation. The newly synthesized RNA was then treated with DNase I (New England Biolabs, Ipswich, MA, USA) to eliminate any residual DNA. The mRNA was purified by precipitation, analyzed using agarose gel electrophoresis, and its concentration was measured photometrically and used in the injection mixture.

The sgRNA spacer sequence targeting exon 51 of the *Dmd* gene was predicted using the Crispor-Tefor [40] online instrument as follows: 5′-TACTCTAGTGACACAATCTG-3′. Two complementary oligonucleotides containing the sgRNA spacer sequence along with overhangs for cloning were annealed and cloned into pSK-T7-Cas9 gRNA in vitro transcription vector treated with BbsI enzyme. A PCR fragment of this vector containing the sgRNA sequence was obtained using Platinum SuperFI high-fidelity polymerase (Thermo Fisher Scientific, Waltham, MA, USA) and used as a matrix for in vitro transcription performed with HiScribe^®^ T7 High Yield RNA Synthesis Kit (New England Biolabs, Ipswich, MA, USA) following the analysis and purification steps described earlier.

### 4.3. Mutant Mouse Generation

The CRISPR/Cas9-based gene editing system was introduced into zygote cytoplasm by microinjection, followed by embryo transfer into pseudopregnant mice to produce genetically modified animals (generation F0). The detailed protocol of zygote collection, culture, microinjection, and transfer into recipients has been previously published by our group [41].

### 4.4. Editing Event Discovery and Genotyping

#### 4.4.1. F0 Generation 

The F0 and F1 generation samples were analyzed by Sanger sequencing (performed by Evrogen, Russia) to verify the presence of editing events. 

Pups’ 3rd toe phalanges of hind limbs collected on postnatal day 7 (P7) for identification purposes were used for analysis. The samples were lysed in an alkaline buffer (25 mM NaOH, 0.2 mM EDTA) at 95 °C for 90 min. The lysates were diluted 50-fold with nuclease-free water, and 1 μL of diluted lysates was directly used as a PCR template in a Phire Tissue Direct (Thermo Fisher Scientific, Waltham, MA, USA) reaction containing 400 nM on-target primers: 5′-TGTTCTCTGGTGTACTGCCT-3′ and 5′-CTCGGTTGAAGTCTGCCAGT-3′. Following the PCR, products were separated in agarose gel. The amplicon bands were then extracted from agarose gel using Monarch DNA Gel Extraction Kit (New England Biolabs, Ipswich, MA, USA) and sequenced.

Sequence traces were analyzed using ICE Synthego (Synthego Performance Analysis, ICE Analysis. 2019. v3.0. Synthego; [accessed on 5 June 2022]), and the editing events were then verified manually based on the traces.

#### 4.4.2. Genotyping

Subsequent generations were analyzed by TaqMan^®^-based real-time PCR assay. 

The following combination of primers and locked nucleic acid probes (LNA probes) was developed and synthesized: Dmd rtgt F (5′-AACACTAGCTGCCAGTCAGAC-3′), Dmd rtgt R (5′-GTAAGTTCTGTCCAAGCTCGG-3′), Dmd insT ROX (ROX-AGTGACACA[+A]TT[+C]TGTGGTTACTA-BHQ2), Dmd insG ROX (ROX-TGACACA[+A]TG[+C]TGTGGTTACT-BHQ2), and Dmd norm FAM (FAM-AGTGACACA[+A]T[+C]TGTGGTTAC-BHQ1), where “[+N]” stands for a locked nucleotide. The two “mutant”-targeted probes were labeled with the same fluorophore, as they were never used together in a mixture to prevent non-specific probe binding or competition.

qPCR was set up in standard 20 μL reactions, containing 400 nM primers and 100 nM probes. Water-diluted lysates (100-fold, 3 μL), obtained as described earlier, were used as PCR templates.

### 4.5. Tissue Preparation

To assess the timing and the course of pathology debut and development, skeletal, intercostal, and cardiac muscles and diaphragm samples were collected from male mice every 7 days starting from P7 up to 6 months of age for the insT line and up to natural death in insG. Histological and statistical analyses are presented on four groups of animals of 2 months old, 25 males in each: insG line, insT line, and two control groups of their wild-type siblings.

After intraperitoneal sedation using Vezotil (VETSTEM^®^ pharma & cell, Moscow, Russia) and Xyla (Interchemie werken “De Adelaar” B.V., Waalre, The Netherlands), transcardiac perfusion with saline followed by 10% neutral buffered formalin (NBF) was performed [42]. Samples of limb muscles, intercostal muscles, hearts, and diaphragms were collected and placed into NBF for 24 h. After the fixation, samples were processed via isopropanol dehydrating solution and mineral oil to paraffin wax according to the manufacturer’s protocol (Medix, Taganrog, Russia). Tissue sectioning was performed using a rotary microtome RMD-3000 (Medtehnikapoint, Saint Petersburg, Russia). Paraffin sections (5 μm) were prepared for both H&E staining and immunofluorescent staining. For H&E staining, Mayer’s Hematoxylin and Eosin aqueous solutions (1%) were used (BioVitrum, Saint Petersburg, Russia). Regaud’s method for identification of early myocardial damage (BioVitrum, Russia, see Appendix A for the protocol) and Haematoxylin-Basic Fuchsin-picric acid (HBFP) staining (BioVitrum, Russia) were used for the early detection of myocardial damage and necrosis. Images were captured using a Nikon ECLIPSE Ti (Nikon Corporation, Tokyo, Japan).

Antibodies used for IHC-P were as follows: Dystrophin, N-terminal (Dystrophin B, NCL-DYSB, Leica Biosystems, Nussloch, Germany), Dystrophin, C-terminal (Rabbit Polyclonal Dystrophin antibody, ab15277, Abcam, Cambridge, UK). Secondary antibody used was Anti-mouse IgG (H+L), F(ab′) 2 Fragment (#4408 Alexa Fluor ^®^ 488 Conjugate, Cell Signaling Technology, Danvers, MA, USA). Nuclear staining used Hoechst 33342 (Thermo Scientific, Waltham, MA, USA).

The blocking solution was formulated as 1% BSA, 0.3% Triton X-100 in 1×PBS. Images were captured using a Leica STELLARIS (Leica Microsystems, Wetzlar, Germany).

### 4.6. Dystrophin Isoform Expression Quantification (qPCR)

For expression quantification, 2- to 3-month-old male mice were used. Total RNA of whole brains, skeletal muscles (gastrocnemius, calf), diaphragms, and intercostal muscles was obtained by phenol-chloroform extraction with ExtractRNA reagent (Evrogen, Moscow, Russia). Diaphragms were promptly rinsed with cold PBS to remove the remaining blood before the extraction. RNA samples (1.5 ug per sample) were treated with MMLV Revertase (Evrogen, Moscow, Russia). Random hexamer primers were used for the first strand cDNA synthesis reaction. The qPCR assay was set up in 384-well plates in three technical replicates for each organ and target (Table 5) using SYBR-green as the detection method. An amount of cDNA sample equivalent to 60 ng of RNA treated was used per replicate. qPCR data were acquired by QuantStudio™ 6 Flex system (Applied Biosystems, Waltham, MA, USA). Expressions were calculated relative to *Hprt* expression. Threshold cycles (Ct) were geometrically averaged over replicates, and the expression level was calculated as follows:$$expr=2^{\bar{Ct}-\bar{Ct(hprt)}}.$$

### 4.7. Statistical Analysis and Data

Statistical analyses were performed using GraphPad Prism 8.0 software using *t*-test, Kruskal–Wallis test, Mann–Whitney test, and Kaplan–Meier estimator. Threshold *p*-values were adjusted to account for multiple comparisons where appropriate. ImageJ Fiji 2.15.1 tools were used to measure fiber Feret diameters and diaphragm widths. Data manipulation was performed in Microsoft Office 2019 Excel.

Wild-type siblings of both mutant groups were post hoc pooled in analyses for plot and table readability purposes, as no difference between them was observed by any statistical test.

## Figures and Tables

**Figure 1 ijms-26-00158-f001:**
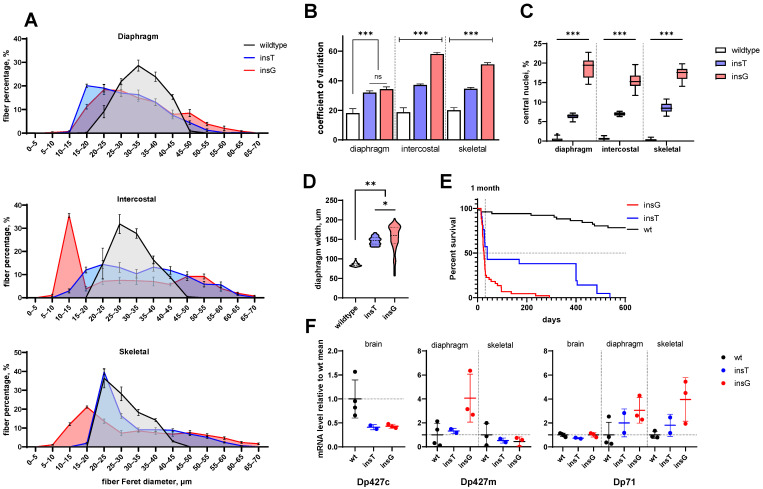
Overall statistics of histopathology, lifespan, and *Dmd* expression in mutant lines (insT, blue and insG, red) and wild-type (wt, black/white) animals. (**A**) Frequency distributions of minimal Feret diameter of fibers in the three corresponding muscular tissues, error bars—95% CI, (**B**) mean coefficient of variation of minimal Feret’s diameter in different tissues, %, error bars—sd, (**C**) percentage of fibers with centrally located nuclei in sections of corresponding muscle tissues (five random animals per group × all nuclei in five sections), (**D**) diaphragm width (five sections per 25 animals in each set), (**E**) survival curves of mutant lines, (**F**) foldchange of dystrophin isoform mRNA level relative to level in wild-type animals. Wild-type sibling animals were pooled to form a single group in all panels. * *p* < 0.03, ** *p* < 0.002, *** *p* < 0.0002.

**Figure 2 ijms-26-00158-f002:**
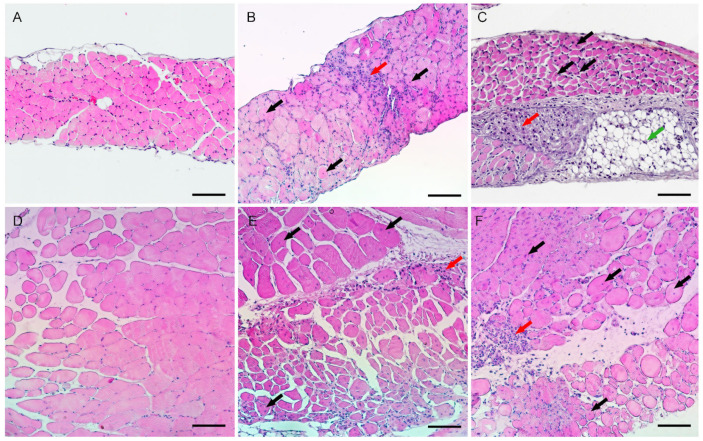
Cross-sections of diaphragms (**A**–**C**) and intercostal muscles (**D**–**F**), H&E staining. (**A**,**D**)—wild type mouse; (**B**,**E**)—insT mutant mouse; (**C**,**F**)—insG mutant mouse. Black arrows—central nuclei; red arrows—fibrosis and necrotic muscle fibers; green arrow—adipose tissue. Scale bar: 100 μm.

**Figure 3 ijms-26-00158-f003:**
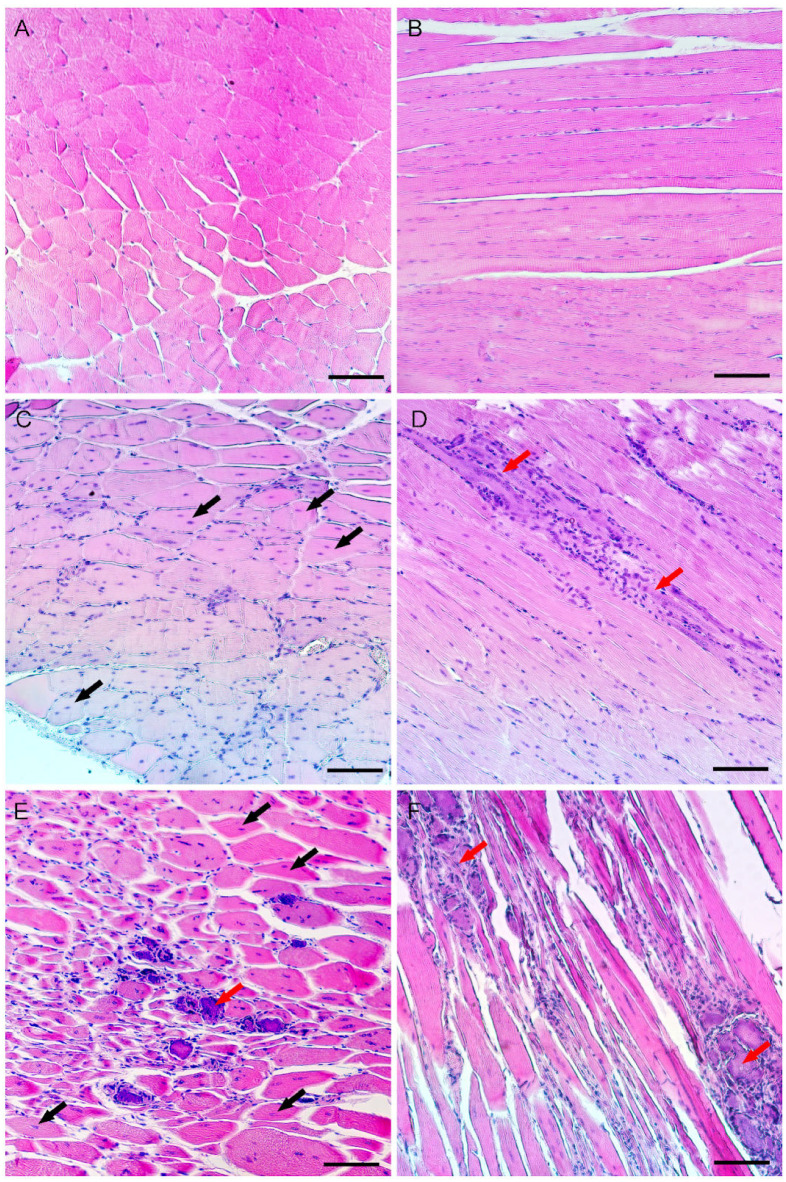
Cross-sections and longitudinal sections of skeletal muscles (gastrocnemius muscles as an example), H&E staining. (**A**,**B**)—wild type mouse; (**C**,**D**)—insT mutant mouse; (**E**,**F**)—insG mutant mouse. Black arrows—central nuclei; red arrows—fibrosis and necrotic muscle fibers. Scale bar: 100 μm.

**Figure 4 ijms-26-00158-f004:**
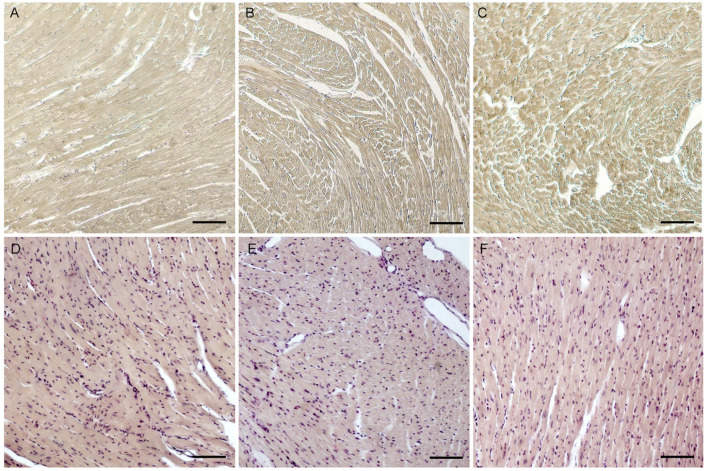
Longitudinal sections of myocardium. (**A**,**D**)—wild-type mouse. (**B**,**E**)—insT mutant mouse. (**C**,**F**)—insG mutant mouse. (**A**–**C**)—Regaud’s iron hematoxylin staining, (**D**–**F**)—HBFP staining. Scale bar: 100 μm.

**Figure 5 ijms-26-00158-f005:**
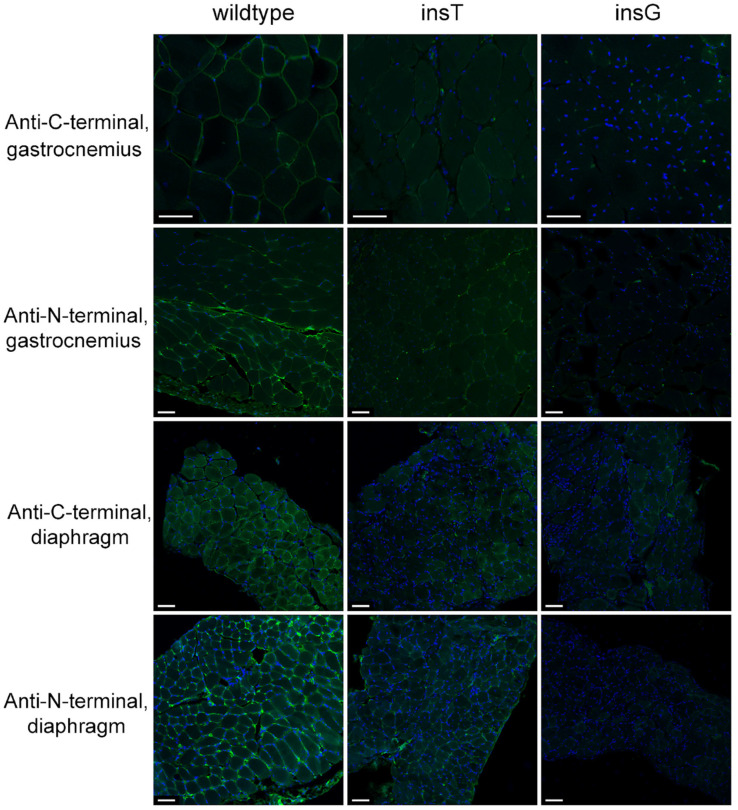
Anti-N-terminal and anti-C-terminal dystrophin immunofluorescence staining on Formalin-Fixed Paraffin-Embedded tissues, in combination with Alexa Fluor^®^ 488. Nuclear staining: Hoechst 33342. Scale bar: 50 μm.

**Table 1 ijms-26-00158-t001:** Quantitative histopathological markers observed in skeletal (Sk) muscle and diaphragms (DIA) of young pups of mutant lines.

Histopathological Markers	insT Line, 3 Weeks Old	insG Line, 1 Week Old
Centrally located nuclei, %	Sk: 2.44 ± 0.64%	Sk: 3.41 ± 1.12%
	DIA: 0.20 ± 0.16%	DIA: 0.96 ± 0.20%
Atrophic muscle fibers and/or myotubules (<20 μm), %	Sk: 3.9 ± 3.6%	Sk: 35.2 ± 2.8%
	DIA: 3.9 ± 3.1%	DIA: 19.1 ± 3.9%
Hypertrophic fibers (>45 μm), %	Sk: none	Sk: 8.4 ± 0.7%
	DIA: 0.2 ± 0.3%	DIA: 2.1 ± 3.6%
Minimal Feret’s diameter coefficient of variation, %	Sk: 18.1%	Sk: 43.8%
	DIA: 21.1%	DIA: 29.8%

Data are presented as mean ± SD. N = 3–4, >150 diameter measurements per N.

**Table 2 ijms-26-00158-t002:** Minimal Feret’s diameter of muscle fibers of generated model animals (insT and insG) and wt = wild type.

	Diaphragm			Intercostal			Skeletal		
	wt	insT	insG	wt	insT	insG	wt	insT	insG
median of minimal Feret’s diameter ± SEM, μm	33.33 ± 0.14	27.32 ± 0.28 ^w,m^	30.57 ± 0.11 ^w,m^	30.4 ± 0.28	33.32 ± 0.43 ^w,m^	27.25 ± 0.35 ^w,m^	26.54 ± 0.26	26.99 ± 0.13 ^w^	26.81 ± 0.29 ^w^
coefficient of variation, %	18.2	31.8 ^w,m^	34.2 ^w,m^	19.1	36.9 ^w,m^	57.8 ^w,m^	20.0	34.5 ^w,m^	50.9 ^w,m^

Data in the first row are presented as mean of medians ± standard error of mean (SEM). Approximately 100 random fibers were measured in mouse (N = 10–15 per group) tissue sections to obtain medians. In the second row, data are presented as mean of coefficients of variation, calculated as a ratio of standard deviation of fiber diameter divided by its mean (~100 fibers for each of 10–15 animals). Superscripts denote significant difference (*p* < 0.05) compared to wild type (w), or between mutants (m).

**Table 3 ijms-26-00158-t003:** Summary of histopathological markers examined in mutant lines generated in this work in skeletal (Sk), intercostal (IC) muscles, and diaphragm (DIA).

Histopathological Markers	insT Line	insG Line
Centrally located nuclei, %	Sk: 8.6 ± 1.2%	Sk: 17.3 ± 1.8%
	IC: 6.9 ± 0.4%	IC: 15.7 ± 2.1%
	DIA: 6.3 ± 0.6%	DIA: 18.8 ± 2.5%
Atrophic muscle fibers and/or myotubules (<20 μm)	Sk: 1.9 ± 0.9%	Sk: 33.8 ± 1.4%
	IC: 14.9 ± 0.8%	IC: 40.2 ± 1.3%
	DIA: 20.6 ± 0.95%	DIA: 12.2 ± 1.36%
Hypertrophic fibers (>50 μm)	Sk: 8.1 ± 1.1%	Sk: 15.9 ± 1.0%
	IC: 12.9 ± 1.1%	IC: 16.1 ± 0.9%
	DIA: 1.8 ± 0.5%	DIA: 6.9 ± 1.0%
Necrosis areas	Individual muscle fibers	From local foci to extensive regions
Endomysium hypertrophy and fibrosis	Local hypertrophy foci	Local hypertrophy and fibrosis foci
Loss of cross-striation	Individual fibers in skeletal and intercostal muscles	Individual fibers in diaphragm, foci in intercostal and skeletal muscles
Macrophage infiltration	Local foci	Spread infiltration foci
Adipose replacement	none	Foci of adipose replacement in diaphragm and intercostal muscles
Myocardial histopathology	none	none

Numerical data are presented as mean ± SD. N ≥ 23 animals per group for qualitative analyses, N = 10–15 for quantitative.

**Table 4 ijms-26-00158-t004:** Comparison of mutant lines generated in this work with some already existing murine models.

Mouse Line/ Characteristics		insT	insG	Mdx (C57BL/10ScSn-Dmd^mdx^/J)	*mdx-utrn^−/−^*	mdx52	DMD-Null
Mutation		Dmd gene exon 51: NM_007868.6:c.7321_7322insT	Dmd gene exon 51: NM_007868.6:c.7321_7322insG	Spontaneous stop codon (3185 C>T conversion) in exon 23	Double knockout Mdx and Utrntm1Ked (Utrn−/−) (neomycin cassette in utrophin exon 7)	Neomycin cassette in exon 52	DMD deletion using Cre-loxP system
Protein		Dp427 absence	Dp427 absence	Dp427 absence	Dp427 and utrophin absence	Dp427, Dp260, and Dp140 absence	Dp427, Dp260, and Dp140 absence
Skeletal muscle and diaphragm pathology	Necrosis/regeneration cycle onset	3–4 weeks old, then stabilization	1.5 weeks old, then pathology development	6 weeks old, then stabilization	1 week old, then pathology development	3–4 weeks old	3–4 weeks old
	Central nucleation	+	+	+	+	+	+
	Fibrosis	Focal	From local to extensive	Usually local	Extensive	Focal (analysis is difficult due to calcification)	Usually local
	Adipose replacement	−	Focal, more common in the diaphragm	Single cases in the diaphragm (12 months and older)	From focal to extensive	Absent or minimal	Rarely, single foci
Cardiopathology		−	−	Subtle cardiomyopathy from 6 months of age	Cardiomyopathy from 8 weeks of age, myocardium fibrosis, left ventricular dilation, cardiac dysfunction	−	Cardiomyopathy from 6 months of age, myocardial fibrosis
Life expectancy		More than 12 months	7 weeks	21–23 months	20 weeks	More than 12 months	More than 12 months
Reference		This work	This work	[39]	[38]	[17]	[15]

The symbols “+” and “−” represent presence or absence of corresponding characteristics, respectively.

**Table 5 ijms-26-00158-t005:** List of primers used for qPCR expression analysis.

Target/Direction	Sequence (5′-3′)	Reference
Dp427m F	AGGAGAAAGATGCTGTTTTGCG	[43]
Dp427m R	AATTGTGCATTTATCCATTTTGTGA	
Dp427c F	AGGAGAAAGATGCTGTTTTGCG	[44]
Dp427c R	AATTGTGCATTTATCCATTTTGTGA	
Dp71 F	ACTCCTCCGCTCTAAGCGT	[45] (a mismatch to the reference genome was fixed)
Dp71 R	CTTCTGGAGCCTTCTGAGC	
*Hprt* F	CAGCGTCGTGATTAGCGATGA	[46]
*Hprt* R	GCCACAATGTGATGGCCTCC	

## Data Availability

The data supporting the findings of this study are available within the article and its Appendix A. Model animals are available upon reasonable request to the corresponding author.

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
