# Peer review of "Two Novel Mouse Models of Duchenne Muscular Dystrophy with Similar Dmd Exon 51 Frameshift Mutations and Varied Phenotype Severity"

_ijms, 2024, doi:10.3390/ijms26010158_

Round 1

Reviewer 1 Report

Comments and Suggestions for Authors

I This study attempts to address the issue for need of mouse model of DMD that better represents the genetic and phenotypic profile of the more common mutations that give rise to human DMD.  This aim of the project is warranted as it well known the limitations of the current mouse models of DMD. They claim that they have generated two new lines of the mouse DMD by Crispr mediated modification of exon 51. 

General Comments:

The project requires significant work in generating the genetically modified mice.  However, the presentation and evidence for genetic modification of the mice is weak. The authors do not show molecular analysis of the mice as they were generated. 

The generation of the Crispr mutant mice is intricate work.  However, the authors do not present any data on the molecular characterization and confirmation of the genetically developed mice.  Definitive proof that the mice have been correctly genetically modified should be provided i.e. PCR confirmation of mutations in each mouse model.  Furthermore, tissues from mouse models should be analysed by immunoblotting with with N- and C-terminus anti-dystrophin antibodies.  The immunofluorescence analysis of muscle sections with the same antibodies is not sufficient. The immunoblotting analysis is especially interesting given the variation of mRNAs for different splice variants of dystrophin in the two mutant mice. 

Specific Comments:

Abstract:

Line 22: List and mention the mouse lines.

Line 27: “….one of the lines.” Indicate the line name. 

Line 137: “….not detected…………”. “However, multiple foci……” New sentence, capital ‘H’.

Line 247: Correct “…. striated muscle…”

Line 250: Modify   Ca2+

Line 290-292: “Mdx52 mouse…….” cite a reference/s to support statement. 

Fig.4:  Immunofluorescent staining is not sufficient in detecting CNFs, NMJs Figure 4.  Add immunblots for molecular confirmation of the mice genotypes.  

Author Response

Comment 1: The project requires significant work in generating the genetically modified mice.  However, the presentation and evidence for genetic modification of the mice is weak. The authors do not show molecular analysis of the mice as they were generated. 

The generation of the Crispr mutant mice is intricate work.  However, the authors do not present any data on the molecular characterization and confirmation of the genetically developed mice.  Definitive proof that the mice have been correctly genetically modified should be provided i.e. PCR confirmation of mutations in each mouse model.

Response 1: Thank you for pointing this out. Indeed, although F0 and F1 generation Sanger sequencing analysis was stated in the Results chapter along with precise characterization of mutation occurring in exon 51, and using molecular LNA probes for genotyping, the presentation of data, proving this, has been omitted.
To definitively prove mutation presence to the readers, Sanger sequence traces of F0 lines’ founder mice and lines’ descendants’ traces are now available in Supplementary materials with a link in section “2.1. Two distinct modified lines were generated”, line 113 .

Comment 2: Furthermore, tissues from mouse models should be analysed by immunoblotting with with N- and C-terminus anti-dystrophin antibodies.  The immunofluorescence analysis of muscle sections with the same antibodies is not sufficient. The immunoblotting analysis is especially interesting given the variation of mRNAs for different splice variants of dystrophin in the two mutant mice.

Response 2: We agree that such protein detection technique as immunoblotting is more robust, reliable and discriminative in many cases, than immunofluorescence. Blotting would also allow us to distinguish between N-terminally stained isoforms, and compare their enrichment. Despite this, Western blotting results were not (yet) yielded due to a few reasons. Firstly, it was not the precise quantification of total cellular isoform diversity, an aim of our work, but rather showing, that DMD loses its membrane localization in our models, proposing the dystrophy observed is associated with its loss, which is detected by immunofluorescence. Secondly, reliably detecting (and quantifying) large proteins, such as 427 kDa isoform of DMD, or its truncated variants, is challenging, mainly at the protein transfer stage. We made attempts to perform Western blotting consecutively. Alas, the results yielded were unsatisfying. As we plan to further elucidate molecular pathogenesis in our apparently dystrophic model strains, this work is of course in our future scope.

Comment 3:

Line 22: List and mention the mouse lines.
Line 27: “….one of the lines.” Indicate the line name. 
Line 137: “….not detected…………”. “However, multiple foci……” New sentence, capital ‘H’.
Line 247: Correct “…. striated muscle…”
Line 250: Modify   Ca2+
Line 290-292: “Mdx52 mouse…….” cite a reference/s to support statement. 

Response 3:

All remarks were considered, and the text and references were modified in corresponding places.

Reviewer 2 Report

Comments and Suggestions for Authors

This work is potentially of significant interest because it indicates (for the first time, to my knowledge) that the loss of dystrophin (and of no other proteins, such as utrophin) in the mouse can result in a severe life-shortening phenotype. It is currently understood in the field that the mouse copes 'well' with loss of dystrophin, and indeed lifespans of existing DMD murine models are close to normal (although the underlying pathology could be considered severe, featuring recurrent bouts of myofibre necrosis and regeneration). The authors' insG model, with its lifespan of just 7 weeks, suggests that this is not always the case. This warrants further investigation.

An important question the authors could try to address is whether the insG line has acquired additional disease-modifying mutations, despite that it has been generated in the same way as their insT line (which does not show a greatly reduced lifespan). For example, comparative sequencing could be performed on both the insT and insG lines to establish if there could be an identifiable genetic basis for the difference between the two lines, and/or sequencing could be performed on some or all of the known modifiers of the DMD phenotype (https://pmc.ncbi.nlm.nih.gov/articles/PMC4591871/). Or perhaps the authors could address this point in some other way. It is important that this question is either addressed directly, or at least discussed in detail in the manuscript, so that the field can understand the implications of the work and the limitations around what we can presently say about these two new models.

If a disease-modifying mutation is not excluded and also not identified then this needs to be clearly stated as a limitation of the work, as it calls into question the utility of the insG model.

If this aspect of the work - i.e. that they seem to have generated a single-mutation DMD murine model with severely reduced lifespan - were to be confirmed and investigated in some depth, then the authors could consider making a bigger point of it (e.g. to mention it directly in the title/abstract).

Minor points:

The introduction should mention existing DMD murine models that are suitable for testing of splicing regulation, and should also mention other DMD murine models and explain why they are not suitable for testing of splicing regulation (some of these are listed in Table 3, but should be introduced in detail in the introduction as well). The mdx52 comes to mind, which has been used to test exon skipping (e.g. https://pubmed.ncbi.nlm.nih.gov/22869723/). Several other models that coud be mentioned include mdx2cv, mdx3cv, mdx4cv, and mdx5cv, mdxBgeo and Dmd-null (see for example https://www.nature.com/articles/s41536-018-0045-4). Currently the introduciton suggests that no models currently exist to test splicing regulation, which is incorrect.

l124-131: "The onset of muscle pathology (central nucleation, pseudohypertrophy) in the insT line occurs at no later than 3 weeks of ....."

Do the authors have any quantitative results to support this difference in timing of pathology onset between the two lines?

Table 1: The formatting shows some of the SEMs on a different row - this makes the table hard to read.

Lastly, it might be useful to indicate if and how the lines could be available to other researchers in the field worldwide, if the authors have a mechanism for this - in the hopes that science succeeds in the present circumstances.

Author Response

Comment 1: An important question the authors could try to address is whether the insG line has acquired additional disease-modifying mutations, despite that it has been generated in the same way as their insT line (which does not show a greatly reduced lifespan). For example, comparative sequencing could be performed on both the insT and insG lines to establish if there could be an identifiable genetic basis for the difference between the two lines, and/or sequencing could be performed on some or all of the known modifiers of the DMD phenotype (https://pmc.ncbi.nlm.nih.gov/articles/PMC4591871/). Or perhaps the authors could address this point in some other way. It is important that this question is either addressed directly, or at least discussed in detail in the manuscript, so that the field can understand the implications of the work and the limitations around what we can presently say about these two new models. If a disease-modifying mutation is not excluded and also not identified then this needs to be clearly stated as a limitation of the work, as it calls into question the utility of the insG model.

Response 1: Female mice of insT and insG lines were back-crossed with wild-type C57Bl/6 males to eliminate potential unwanted off-target action of CRISPR/Cas9 system from genome, as target mutations were selected by genotyping, while any other mutations that could have occurred in some distance from Dmd gene — weren’t. Male animals of generations F2–F4 were included in experimental samples — we compared mutant subjects and their wild-type siblings, that would also carry unwanted off-target DMD-associated mutations should there be any present in their mutant parent’s genome. To prove there are no other mutations in the Dmd ORF, sequencing would indeed be of high interest, but has not been performed as of now due to complexity of such procedure on a long cDNA and possible nonsense-mediated decay in mutant lines. In chapter 2.1, lines 115–117 were added to state this. Lines were added to the last paragraph of Discussion chapter.

Comment 2: The introduction should mention existing DMD murine models that are suitable for testing of splicing regulation, and should also mention other DMD murine models and explain why they are not suitable for testing of splicing regulation (some of these are listed in Table 3, but should be introduced in detail in the introduction as well). The mdx52 comes to mind, which has been used to test exon skipping (e.g. https://pubmed.ncbi.nlm.nih.gov/22869723/). Several other models that coud be mentioned include mdx2cv, mdx3cv, mdx4cv, and mdx5cv, mdxBgeo and Dmd-null (see for example https://www.nature.com/articles/s41536-018-0045-4). Currently the introduciton suggests that no models currently exist to test splicing regulation, which is incorrect.

Response 2: Thank you for bringing this forward and suggesting the models to mention. The introduction has been expanded (lines 65-97) to better shade light on currently existing models and their limitations.

Comment 3: l124-131: "The onset of muscle pathology (central nucleation, pseudohypertrophy) in the insT line occurs at no later than 3 weeks of ....." Do the authors have any quantitative results to support this difference in timing of pathology onset between the two lines?

Response 3: A new table is now added, numbered as “Table 1”, containing quantitative description of few young pups of the mutant lines. A Figure S2 has been added to Supplementary Materials, revealing pathological changes observed in young pups.

Comment 4: Table 1: The formatting shows some of the SEMs on a different row - this makes the table hard to read.

Response 4: The table, now being numbered “Table 2”, is modified for better readability in the draft.

Comment 5: Lastly, it might be useful to indicate if and how the lines could be available to other researchers in the field worldwide, if the authors have a mechanism for this - in the hopes that science succeeds in the present circumstances.

Response 5: We are open to collaboration and sharing our work. The best that could be done, to our mind, is to put a strain upon-request availability statement in the backmatter, although this has to be discussed with the editorial board.

Reviewer 3 Report

Comments and Suggestions for Authors

The authors generate and describe two genetically modified mouse lines (InsT and InsG) that are suitable models of DMD suggesting that they can be used to test gene therapy based on alternative splicing.

The manuscript is well written and the results are well described and discussed.

Although my opinion of this study is very positive, I think it is necessary to add 2 measures:

1) western blot of dystrophin in all experimental models. This analysis should be combined with immunofluorescence as it is more specific and quantifiable. Immunofluorescence alone is not enough.

2) in vivo measurements of muscle strength and contraction capacity are needed in all experimental models.

Author Response

Comment 1: The authors generate and describe two genetically modified mouse lines (InsT and InsG) that are suitable models of DMD suggesting that they can be used to test gene therapy based on alternative splicing. The manuscript is well written and the results are well described and discussed. Although my opinion of this study is very positive, I think it is necessary to add 2 measures: 1) western blot of dystrophin in all experimental models. This analysis should be combined with immunofluorescence as it is more specific and quantifiable. Immunofluorescence alone is not enough.

Response 1: We are very obliged to hear this. Thank you for such appreciation of our work. On the case of western blotting, the main purpose of our work was to justify the loss of membrane localization of the DMD in muscular tissues associated with observed dystrophy. We agree, that western blotting technique is more robust, discriminative and quantifiable. But such an experiment, to our belief, is redundant to suggest that the models bear C-terminally-impaired DMD variants and the full isoform has substantially lost its membrane localization in the insG line, while being necessary to further investigate molecular pathology. Additionally, running Western blot on large proteins reliably is not easily feasible.

Comment 2: 2) in vivo measurements of muscle strength and contraction capacity are needed in all experimental models.

Response 2: Agreed, such functionality assessments of motor functions would be of great value to prove that our models mimic human pathology in essence and is of big relevance to whether our models would be confirmed as suitable for testing functionality-restoring therapies. Nevertheless, while there are still many uncharacterized aspects of our models, which require by long chalks more work, we believe that they are already of big significance due to the strong phenotype of insG line and its substantially reduced lifespan, never observed in Dmd-mutation models. The last paragraph of Discussion section was modified, stating this as limitation.

Round 2

Reviewer 1 Report

Comments and Suggestions for Authors

I am generally satisfied with the revised manuscript, except do the authors have explanation for why they do not see a cardiac phenotype/dystrophy in their new mouse models of DMD as this is in contrast to most DMD animal models ?

Author Response

Comment: I am generally satisfied with the revised manuscript, except do the authors have explanation for why they do not see a cardiac phenotype/dystrophy in their new mouse models of DMD as this is in contrast to most DMD animal models ?

Response: Thank you for your comment. To our knowledge, not so many murine DMD models exhibit pronounced cardiopathology, but those, that do, either:
1) have its onset at later stages of their lives (more than 6 months of age), and our insG animals are mostly extinct by this time, and insT were not examined at this age for heart damage,
or 2) require more than a single mutation close to the protein's C-terminus - either full or substantial loss of DMD, or Utrophin, or multi-exonic mutations. 
We examined the state of our models at the age of 2 to 3 months.
To make this more apparent to the readers, we now speculate on it in lines 339-346.

Reviewer 2 Report

Comments and Suggestions for Authors

I appreciate the additions the authors have made to the text, which address my concerns in the introduction, about the variety of existing murine DMD models, and in the methods and discussion, to clarify that, although back-crossing has been performed, a disease-modifying mutation has not been ruled out, and that further work is needed to understand the reason for the difference in severity between the two models.

The main text of the paper is now more accurate and is supported by the results.

However, with this clarification, it is now apparent that the title is misleading and likely incorrect, as we do not know if it is the exon 51 frameshifts that have induced the difference in severity (and this is likely not to be the case). The title should be reformulated to remove the word "Induce". For example, it could read something like: "Two new mouse models of DMD, carrying exon 51 frameshift mutations homologous to those found in human patients, and having different phenotype severities".

The abstract also is somewhat misleading without some form of qualification of the final sentence. It would be better if the abstract explains that further work is needed to determine the basis of the difference in severity. For example, the last sentence could read something like, "Both genetically modified mouse lines are suitable models of DMD and can be used to test gene therapy based on alternative splicing, although further work is needed to determine the basis, such as a disease-modifying mutation, for the difference in severity between the two models."

Minor point:

- Abbreviations in table 1 (e.g. 'sk', 'DIA') should be explained

Author Response

Comment 1: However, with this clarification, it is now apparent that the title is misleading and likely incorrect, as we do not know if it is the exon 51 frameshifts that have induced the difference in severity (and this is likely not to be the case). The title should be reformulated to remove the word "Induce". For example, it could read something like: "Two new mouse models of DMD, carrying exon 51 frameshift mutations homologous to those found in human patients, and having different phenotype severities".

Response 1: Thank your for your proposition. Of course, the title must be changed now so as not to mislead readers. The new title goes like: "Two Novel Mouse Models of Duchenne Muscular Dystrophy with Similar Dmd Exon 51 Frameshift Mutations and Varied Phenotype Severity". We decided to omit "found in human patients" as this would just make the title too long.

Comment 2: The abstract also is somewhat misleading without some form of qualification of the final sentence. It would be better if the abstract explains that further work is needed to determine the basis of the difference in severity. For example, the last sentence could read something like, "Both genetically modified mouse lines are suitable models of DMD and can be used to test gene therapy based on alternative splicing, although further work is needed to determine the basis, such as a disease-modifying mutation, for the difference in severity between the two models."

Response 2: Thank you for this point. The abstract now lets the readers forsee limitations and further work in the abstract. Its last sentence was modified (lines 27-29): Although further work is needed to qualify these mutations as sole origins of dissimilarity, both genetically modified mouse lines are suitable models of DMD and can be used to test gene therapy based on alternative splicing

Comment 3: - Abbreviations in table 1 (e.g. 'sk', 'DIA') should be explained

Response 3: Thank you for your attentiveness. Table 1 caption was modified.

Reviewer 3 Report

Comments and Suggestions for Authors

Response 2: Although I still think that in general, a scientific paper in which an animal model is characterized cannot be valid without in vivo analysis, I still accept that this was included as a limitation and suggest that this be remembered for future works.

Response 1: In this case, however, I continue to disagree. Following the proper protocol with slow electrophoresis and longer transferring protein from gel to PVDF, DMD is obtained quite easily, especially on animal tissue. If more tissue is available, I still think that a western blot for the evaluation of dystrophin expression should be done.  

Author Response

Comment 1: Although I still think that in general, a scientific paper in which an animal model is characterized cannot be valid without in vivo analysis, I still accept that this was included as a limitation and suggest that this be remembered for future works.

Response 1: We appreciate your understanding regarding the limitations of our current study and acknowledge your point on the importance of in vivo analysis for comprehensive characterization. We have noted this for our future research endeavors, where we aim to incorporate in vivo analysis to enhance the robustness and reliability of our findings.

Comment 2: In this case, however, I continue to disagree. Following the proper protocol with slow electrophoresis and longer transferring protein from gel to PVDF, DMD is obtained quite easily, especially on animal tissue. If more tissue is available, I still think that a western blot for the evaluation of dystrophin expression should be done. 

Response 2: We couldn't agree more that Western blot is a way to go for a more elaborate investigation on Dystrophin expression in our models. There is one more complication that now limits our work on it currently --- the lack of an appropriate and specific N-terminus-staining antibody for murine DMD. We've tried a few, and found Leica DYSB antibodies (anti-human), which suited us best for IHC, but
1) disappointingly, they don't seem to work for Western blot, as manufacturer confirmed
2) we no longer have them, and obtaining new in current conditions is intricate.
We would be grateful if one could recommend 100% working specific antibodies for murine DMD.